# Adaptation and Application of the large LAERTES-EU RCM Ensemble for Modeling Hydrological Extremes: A pilot study for the Rhine basin

Florian Ehmele[1], Lisa–Ann Kautz[1], Hendrik Feldmann[1], Yi He[2], Martin Kadlec[3], Fanni D. Kelemen[1,4], Hilke S. Lentink[1], Patrick Ludwig[1], Desmond Manful[2], and Joaquim G. Pinto[1]

[1]Institute of Meteorology and Climate Research, Karlsruhe Institute of Technology (KIT), Hermann–von–Helmholtz–Platz 1, 76344 Eggenstein–Leopoldshafen, Germany.
[2]Tyndall Centre for Climate Change Research, School of Environmental Science, University of East Anglia (UEA), Norwich, United Kingdom.
[3]Impact Forecasting, Aon, Prague, Czech Republic.
[4]now at: Institute for Atmospheric and Environmental Sciences, Goethe University Frankfurt, Frankfurt am Main, Germany.

**Correspondence:** Florian Ehmele (florian.ehmele@kit.edu)

**Abstract.** Enduring and extensive heavy precipitation associated with widespread river floods are among the main natural hazards affecting Central Europe. Since such events are characterized by long return periods, it is difficult to adequately quantify their frequency and intensity solely based on the available observations of precipitation. Furthermore, long-term observations are rare, not homogeneous in space and time, and thus not suitable to run hydrological models (HMs) with respect to extremes. To overcome this issue, we make use of the recently introduced LAERTES-EU (**LA**rge **E**nsemble of **R**egional clima**T**e mod**E**l **S**imulations for **EU**rope) data set, which is an ensemble of regional climate model simulations providing over 12,000 simulated years. LAERTES-EU is adapted for the use in an HM to calculate discharges for large river basins by applying a quantile mapping with a parameterized gamma distribution to correct the mainly positive bias in model precipitation. The Rhine basin serves as a pilot area for calibration and validation. The results show clear improvements in the representation of both precipitation (e.g., annual cycle and intensity distributions) and simulated discharges by the HM after the bias correction. Furthermore, the large size of LAERTES-EU improves the statistical representativeness also for high return values above 100 years of discharges. We conclude that the bias-corrected LAERTES-EU data set is generally suitable for hydrological applications and posterior risk analyses. The results of this pilot study will soon be applied to several large river basins in Central Europe.

## 1 Introduction

River (fluvial) floods are among the most disastrous and also costliest weather-related hazards in Central Europe (e.g., Alfieri et al., 2018). The damage caused by the devastating 2013 Elbe and Danube flood in Germany (e.g., Grams et al., 2014; Kelemen et al., 2016) has been estimated at 12 billion Euro (Merz et al., 2014). Major flood events along the main river networks are

generally related to the occurrence of intensive and/or long-lasting, mainly stratiform precipitation (e.g., Maddox et al., 1979; Hilker et al., 2009; Schröter et al., 2015).

Due to the huge impact of flooding on human activities, economy, agriculture, infrastructure, and transport, there is a high interest in quantifying the risk of flooding for Central Europe (e.g., Ward et al., 2011; Feyen et al., 2012; Jongman et al., 2014). Despite the occurrence of several prominent events during the last decades, extreme floods have typically long return periods around or above 100 years (e.g., Pauling and Paeth, 2007; Hirabayashi et al., 2013), and thus only a few events are represented in short-term (observational) data sets. Long-term observational records of precipitation are limited and of heterogeneous quality across Europe. To overcome this shortcoming, observations are usually extrapolated using statistical approaches like fitting various probability density functions to a reduced data series (annual maxima or peak over threshold) which show a rather large uncertainty for high return periods (e.g., Lang et al., 2010; Volpi et al., 2019). Nevertheless, it is expedient to use long-term data sets to run hydrological models (HMs) for proper flood risk estimation of high return periods (e.g., Feyen et al., 2012), such as the one-in-200-years event required by the insurance regulation of Solvency II.

On the other hand, reanalyses products (e.g., Dee et al., 2011) provide homogeneous data sets covering long time periods with the limitation of a comparatively coarse resolution. Approaches to overcome the shortcoming of small sample sizes focus on the development of stochastic precipitation models (e.g., Richardson, 1981; Ehmele and Kunz, 2019) or the downscaling of long-term reanalyses or global climate models (GCMs) by regional climate models (RCMs, e.g., Gutmann et al., 2012; Ott et al., 2013; Stucki et al., 2016). Additionally, combined approaches, so-called statistical–dynamical downscaling methods, are also used (e.g., Fuentes and Heimann, 2000; Reyers et al., 2015). The added value of the high-resolution RCMs compared to GCMs is discussed, for example, in Feser et al. (2011) or Feldmann et al. (2013). One of the key benefits is the better representation of the spatial and intensity distribution of precipitation, which is crucial for hydrological modeling particularly over complex terrain (Frei et al., 2000). However, the spatial resolution of RCMs may still be too coarse to effectively model the hydrological processes essential for quantifying flood risk. Although expected further enhancement in model resolution will undoubtedly improve the representation of precipitation, especially for convective-scale events (e.g., Coppola et al., 2020), significant challenges will remain for the foreseeable future (Cloke et al., 2013).

Furthermore, several challenges remain when producing precipitation statistics that are adequate for climate impact studies regarding flooding (e.g., Teutschbein and Seibert, 2010). First, a bias correction of the simulated precipitation is required (e.g., Berg et al., 2012; Ehret et al., 2012). This necessity arises from the shortcomings of the RCMs, which can result from an imperfect model structure, errors in the parameterization scheme, an incorrect initialization, or they can be inherited from the driving GCM (e.g., Ehret et al., 2012; Chen et al., 2018). Moreover, RCMs generally overestimate precipitation across the distribution spectrum (e.g., Feldmann et al., 2008; Berg et al., 2012). An overview of different bias correction methods for hydrological impact studies can be found in Teutschbein and Seibert (2012) or Teng et al. (2015).

The added value of a bias correction for hydrological modeling has been assessed for example in Chen et al. (2019). They focused on the Hanjiang River in south-central China for the period 1961–2000 and calculated streamflow metrics with a 21-parameter lumped, conceptual, rainfall–runoff model from corrected and uncorrected GCM ensemble data. They concluded

that a bias correction is important to simulate reasonable discharges. However, in other studies (e.g., Chen et al., 2018) the results were mixed.

Many studies have demonstrated the added value of a bias correction for precipitation without any linkage to hydrological applications (e.g., Dobler and Ahrens, 2008; Fang et al., 2015). Dobler and Ahrens (2008) compared different downscaling approaches for precipitation in Europe and South Asia as well as different bias correction methods (quantile mapping and local intensity scaling). The authors concluded that dynamical downscaling with an RCM in combination with a bias correction (quantile mapping with a gamma distribution) is most suitable to simulate precipitation in Europe. Fang et al. (2015) focused on the comparison of different bias correction methods and found that empirical quantile mapping and power transformation performed best for precipitation. However, they mentioned that the selection of an accurate correction method may be case sensitive.

The present study emanates from an interdisciplinary project aiming to quantify the flood risk for large European river basins using a model chain from meteorology over hydrology towards risk assessment. The novel RCM ensemble LAERTES-EU (**LA**rge **E**nsemble of **R**egional clima**T**e mod**E**l **S**imulations for **EU**rope), which was recently introduced by Ehmele et al. (2020), is now adapted and applied for hydrological applications. With this aim, daily precipitation amounts and daily mean 2-meter temperature are used as input data to drive an HM for discharge simulations. Ehmele et al. (2020) identified a positive bias in LAERTES-EU precipitation compared to observations, which would lead to an overestimation of the HM discharge response without a previous bias correction. We elaborate the effects of the bias correction to both precipitation and discharge statistics and demonstrate the benefits of a data set like LAERTES-EU for hydrological applications such as the estimation of extreme discharges with high return periods and their statistical representation. We focus on the Rhine basin as a pilot area and address the following research questions:

1. *Does the bias correction improve the representation of precipitation in LAERTES-EU adequately?*

2. *Is the applied HM capable of reproducing observed historical discharges?*

3. *Does the bias-corrected LAERTES-EU provide the potential to derive statistically robust estimates of flood return levels above 100 years?*

This paper is structured as follows: The used data sets and the study area are introduced in Sect. 2. Section 3 contains the atmospheric part with the description and validation of the bias correction method. In Sect. 4, the hydrological model is introduced and validated. In Sect. 5, the benefit of a data set such as LAERTES-EU for hydrological modeling is demonstrated. The last section (Sect. 6) summarizes the results and provides the conclusions.

## 2 Data sets and study area

This study is based on the LAERTES-EU ensemble of RCM simulations (Ehmele et al., 2020), which is introduced in this section as well as different observational data sets used for calibration and validation of both the HM and the bias correction.

## 2.1 LAERTES-EU

The RCM ensemble LAERTES-EU (Ehmele et al., 2020) was produced within the German national research project (BMBF) *Mittelfristige Klimaprognosen* (MiKlip, Marotzke et al., 2016). The non-hydrostatic COSMO model in its climate mode (COSMO-CLM[1], Consortium for Small-Scale Modeling Climate Limited-area Model, thereafter CCLM; Rockel et al., 2008) was used for dynamical downscaling of global MPI-ESM (Max-Planck-Institute Earth System Model; e.g., Giorgetta et al., 2013; Müller et al., 2018) simulations to a horizontal resolution of $0.22°$ ($\sim 25\,\text{km}$) covering the EURO-CORDEX domain[2].

LAERTES-EU consists of four data blocks (Table 1) distinguishing between different resolutions and initialization of the MPI-ESM global model used as boundary conditions. The RCM (CCLM) version, setup, and initialization method remain the same for all simulations (Feldmann et al., 2019). Data blocks 1 and 2 are forced with the low-resolution (T63, $\approx 200\,\text{km}$) version MPI-ESM-LR, while data blocks 3 and 4 use the high-resolution (T127, $\approx 100\,\text{km}$) MPI-ESM-HR (Müller et al., 2018). For both MPI-ESM resolutions, one data block contains long-term transient simulations (1 and 3) while the other consists of 95 multiple-member decadal (10-year) hind- and forecast simulations. As described in Ehmele et al. (2020), a drizzle and dry-day correction is applied to the LAERTES-EU data set to reduce well-known RCM artifacts (e.g., to much drizzle).

For data block 1, three members of the 20CR reanalysis data (Compo et al., 2011) are assimilated to the MPI-ESM-LR and dynamically downscaled with CCLM providing 110 transient years each (Müller et al., 2015). Data block 3 consists of 5 members forced with so-called historical simulations of MPI-ESM-HR using CMIP5 observed natural and anthropogenic 100 climate forcing (Taylor et al., 2012). Three members cover the time period 1900–2005 (106 years each), the two others cover the years 1960–2005 (46 years each).

In the present study, we focus on the data blocks 2 and 4, which make up approximately 95% of the whole LAERTES-EU data set. Both data blocks consist of decadal simulations which run free after the first initialization. Data block 2 has three members, each with 100 simulated decades. The starting conditions are derived from the transient simulations of data block 105 1. Starting in 1910, all three members simulate a 10-year period. For the next hindcast, the initialization point is shifted by 1 year until the last starting year 2009 (simulation end in 2019). Data block 4 is divided into two parts both covering the time period 1961–2026. The initial conditions for the first part are derived from the MPI-ESM-HR with CMIP5 forcing and includes 5 members. The second part consists of 10 members using initial conditions from the MPI-ESM-HR with CMIP6 forcing (Eyring et al., 2016).

For more details on the forcing data, the performance, the added value of LAERTES-EU in comparison to GCM simulations, as well as on the advantages of the ensemble approach, we refer to Ehmele et al. (2020).

---

[1]http://www.cosmo-model.org; last access: Sept. 2020
[2]http://www.euro-cordex.net; last access: Sept. 2020

**Table 1.** Overview of the RCM ensemble LAERTES-EU with the classification into data blocks, the underlying forcing data, the covered time period, and the number of members and simulation years. Table adapted from Ehmele et al. (2020).

| block | forcing | period | member | years |
|-------|---------|--------|--------|-------|
| 1 | 20CR via MPI-ESM-LR | 1900–2009 | 3 | 330 |
| 2 | MPI-ESM-LR DROUGHTCLIP | 1911–2019 | 3 | 3,000 |
| 3 | MPI-ESM-HR HISTORICAL | 1900–2005 | 5 | 410 |
| 4 | MPI-ESM-HR CMIP5 | 1961–2026 | 5 | 2,850 |
|  | MPI-ESM-HR CMIP6 | 1961–2026 | 10 | 5,700 |

## 2.2 Observational data

### 2.2.1 E-OBS

Observed daily precipitation sums and mean temperature on a 0.22° resolution grid were obtained from the E-OBS data set (v17; Haylock et al., 2008; Van den Besselaar et al., 2011) in consistency to Ehmele et al. (2020). E-OBS is widely used for model validation (e.g., Min et al., 2013) and for climatological studies (e.g., van Oldenborgh et al., 2016). The accuracy of E-OBS depends on the station network density (Cornes et al., 2018), which is not homogeneous across Europe. Moreover, Haylock et al. (2008) pointed out that rainfall totals might be reduced in comparison to the raw station data. Nevertheless, and with respect to the overall aim of a consistent approach for several large European river basins, not only the Rhine, E-OBS is the most suitable reference data for the applied bias correction.

### 2.2.2 HYRAS

To estimate the added-value of the bias correction of precipitation, we consider the high-resolved ($5 \times 5$ km$^2$) HYRAS (**HY**dro-meteorological **RAS**ter) data set provided by the German Weather Service (DWD; Rauthe et al., 2013) as an independent data set. Aggregated to the RCM/E-OBS grid (25 km), HYRAS is used for the validation of the bias correction. In its original resolution, HYRAS is used for the calibration and validation of the HM. Note that HYRAS data are not homogeneous over time due to the changing number, location, and instrumentation of the observations. Furthermore, there is a certain bias in precipitation totals especially over complex terrain, where the number of observations is limited (e.g., Piani et al., 2010; Kunz, 2011; Berg et al., 2012).

### 2.2.3 Discharge observations

For the calibration of the rainfall-runoff model, daily mean values of runoff are required. We have selected 71 gauging stations in the Rhine basin, all of them having at least 20 years of continuous observations. The discharge data have various sources: the major part (40 gauging stations) is provided by the German Federal Institute of Hydrology[3], the rest is operated by the individual state ministries of environment from North Rhine-Westphalia[4], Rhineland-Palatinate[5], Baden-Wuerttemberg[6], Hesse[7], Bavaria[8], and Saarland[9]. Two gauging station have been provided by the Swiss Federal Office for the Environment (FOEN).

## 2.3 Study area and time period

The focus in this study is on the Rhine river basin as a pilot area. The river Rhine has a length of about 1,200 km and a total basin size of approximately 185,000 km$^2$.[10] The annual mean discharge close to the estuary is 2,173 m$^3$ s$^{-1}$ (Tockner et al., 2009; Hein et al., 2019). The source of the Rhine is located in the high Alpine Mountains. The basin is characterized by various terrain with mountains up to 4,000 m in the headwaters, rolling hills with elevations around 1,000 m and below in the middle part, and mostly flat lands in the northern part (Fig. 1a). Furthermore, the study area covers different precipitation climatologies. As shown for example by Ionita (2017), the mean annual precipitation exceeds more than 2,000 mm over a large area of the Rhine spring area. Due to the high elevation, a significant proportion falls as snow, especially in winter. As snow melt can be an important component for HMs (cf. Sect. 4.1), the impact of the terrain is expected to be higher for the Alpine catchments than elsewhere. For the remaining study area, the annual precipitation amounts are generally below 1,000 mm (e.g., Tapia et al., 2015).

The Rhine basin is divided into 71 catchments associated with the same number of gauging stations (cf. Sect. 2.2.3). Out of these 71 stations, we selected 6 for this study with various catchment size (Table 2 and Fig. 1b) to compare the observed and simulated discharges for past flood events.

The investigation period is limited by the given data sets. Using LAERTES-EU data blocks 2 and 4 and HYRAS, we focus on the period 1961–2006 for validation and calibration, which is covered by all precipitation data sets. Regarding the statistical analysis, all available data are taken into account.

## 3 Bias Correction of Precipitation

In this section, we describe and validate the applied bias correction with respect to the statistical representation of precipitation within LAERTES-EU as the method itself has been validated by numerous previous studies (cf. below).

---

[3]Wasser- und Schifffahrtsverwaltung des Bundes (WSV), provided by Bundesanstalt für Gewässerkunde (BfG)

[4]Land NRW; dl-de-by-2.0 (www.govdata.de/dl-de/by-2-0) www.elwasweb.nrw.de

[5]Ministerium für Umwelt, Energie, Ernährung und Forsten Rheinland-Pfalz

[6]Pegeldaten der Landesanstalt für Umwelt, Messungen und Naturschutz Baden-Württemberg (LUBW)

[7]Hessisches Landesamt für Naturschutz, Umwelt und Geologie

[8]Bayerisches Landesamt für Umwelt, www.lfu.bayern.de

[9]Ministerium für Umwelt und Verbraucherschutz Saarland

[10]https://www.eea.europa.eu/archived/archived-content-water-topic/rivers/european-river-catchments; Last access Oct. 2020

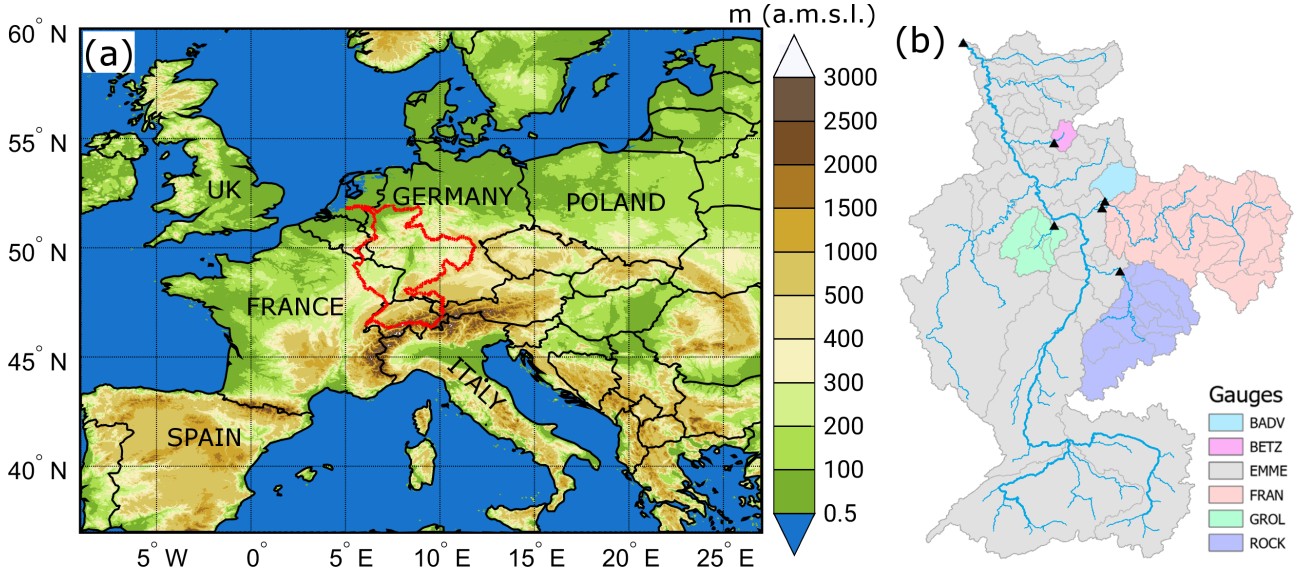

**Figure 1.** Maps of the Rhine basin with (a) the elevation (in meters above mean sea level; basin marked with red contour) and (b) overview of the location (triangles) and associated catchments (colored shading) of gauging stations that were chosen for model validation.

**Table 2.** List of gauging stations (full name, used abbreviation (code), associated river system, and length $L$ of the time series) used for the validation of the hydrological model for selected historical flood events sorted by the upstream catchment size (A).

| Code | Gauge name | River | A [km$^2$] | L [yrs] |
|------|-----------|-------|-----------|---------|
| BETZ | Betzdorf | Sieg | 756 | 63 |
| BADV | Bad Vilbel | Nidda | 1,619 | 57 |
| GROL | Grolsheim | Nahe | 4,012 | 39 |
| ROCK | Rockenau | Neckar | 12,710 | 66 |
| FRAN | Frankfurt Osthafen | Main | 24,764 | 53 |
| EMME | Emmerich | Rhine | 159,555 | 61 |

## 3.1 Quantile mapping technique

Ehmele et al. (2020) showed that LAERTES-EU can produce a reasonable evolution of areal precipitation extremes over Central Europe and the Alpine region for the last century. Although a dry–day correction using E-OBS is already applied, there is still an offset between observations and LAERTES-EU for the considered yearly percentiles of spatial mean precipitation, indicating the need of further post-processing. As a positive bias in precipitation would result in overestimated discharges, a bias correction of LAERTES-EU is inevitable.

The review of Maraun (2016) or the study of Fang et al. (2015) provide a detailed overview of various bias correction methods. The selection of the most suitable method often depends on the application. Nevertheless, the gamma distribution seems to be most suitable in using the quantile method for correcting precipitation. For this study, we therefore use the gamma quantile mapping (GQM) technique with different correction functions for each month. The corrected precipitation amount can be calculated as follows (e.g., Gutjahr and Heinemann, 2013):

$$x_{\mathrm{corr},m,d} = F_{\mathrm{obs},m}^{-1}\left(F_{\mathrm{raw},m}\left(x_{\mathrm{raw},m,d}\right)\right) , \tag{1}$$

where $x$ is the precipitation of either the raw model ("$_{\mathrm{raw}}$"), or the bias corrected model ("$_{\mathrm{corr}}$"); $m$ denotes the month, while $d$ is the day within month $m$. $F$ is the cumulative density function of the gamma distribution, and $F^{-1}$ its inverse with ("$_{\mathrm{obs}}$") referring to the observations.

The applied bias correction aims to improve the intensity of daily precipitation considering each month separately to account for seasonality. Building $F$ both for observed and simulated precipitation, the probability of the model intensities is adjusted to those of the observations. Using a parameterized density function instead of an empirical approach allows to retain the heavy tail of the model distribution to a high degree, which represents the unknown and not yet observed range of intensities. The correction factors for the gamma distributions were defined separately for each data block and month. Therefore, all members within a data block are first concatenated and treated as a single data set to which in a second step a gamma distribution is fitted. We did not correct the individual members independently as such an approach would force all members to the target (observed) distribution which would result in a reduced ensemble spread and thus, an underestimated natural internal climate variability (Chen et al., 2019).

Bias correction methods are statistical approaches and are able to improve mean values and distributions in a way that they become closer to those of the reference data (e.g., White and Toumi, 2013). However, they are not able to improve the simulated precipitation in terms of timing or underlying dynamical processes (Ehret et al., 2012). Another limitation of bias correction is that stationarity of the model bias is assumed (Maraun, 2012; Chen et al., 2015). Furthermore, there are no suitable observations available for the period prior 1950 and the predictions until 2028, for which we also assume stationarity of both model bias and precipitation distribution.

Please note that daily mean 2 m temperature has also been bias-corrected using quantile mapping with a parameterized Gaussian distribution (e.g., Piani et al., 2010). The bias-corrected temperature data is used in line with the bias-corrected precipitation to run the HM (cf. Sect. 4). Nevertheless, the focus of this study remains precipitation as the model uncertainties are higher and it is the dominant factor in case of major flood events.

### 3.2 Validation of bias-corrected precipitation

The bias of the corrected and uncorrected LAERTES-EU data block 2 ensemble mean is shown in Fig. 2. For the uncorrected precipitation, a positive bias is visible within almost the entire Rhine basin compared to E-OBS and HYRAS (Fig. 2a,c). Overall, a clear improvement is found after bias correction (Fig. 2b). The remaining precipitation bias relative to E-OBS is mostly positive but below 0.2 mm. The residual differences are higher up to 0.4 mm only in the southernmost part of the

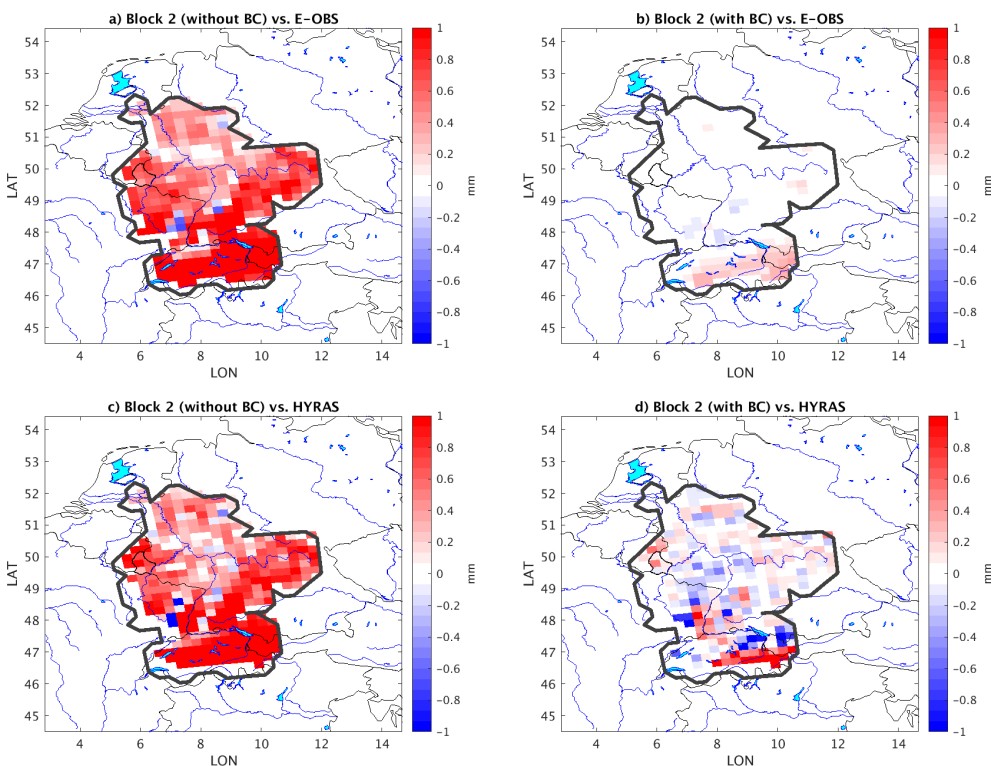

**Figure 2.** Bias within in the Rhine basin (bold black contour) of daily precipitation [in mm] for the LAERTES-EU ensemble mean based on data block 2 (a) towards E-OBS uncorrected, (b) towards E-OBS after bias correction, and (c), (d) towards HYRAS, respectively. Thin black lines show country borders, blue lines indicate rivers, and cyan shaded areas show lakes.

Rhine basin. As part of the Alpine Mountains, this area is characterized by complex topography and high spatial variability of
195 precipitation which is difficult to capture for the RCM. Furthermore, we do not bias-correct each member of LAERTES-EU separately but block-wise in order to preserve the internal ensemble variability which affects the ensemble mean as well. The small residual biases were expected because E-OBS was used as the training data in the bias correction.

A strong reduction of the bias is also shown when comparing LAERTES-EU with HYRAS. While the uncorrected model precipitation is overestimated compared to the observed precipitation in HYRAS (Fig. 2c), the bias correction clearly reduces
this overestimation. This results in a slight under-representation of rainfall at most grid points (Fig. 2d). The mainly negative differences in the corrected model data towards HYRAS derives from the differences between HYRAS and E-OBS, since E-OBS itself shows a negative precipitation bias (e.g., Haylock et al., 2008). Again, the highest values are found in the Alpine region. Similar results can be found for LAERTES-EU data block 4 (see Fig. S1 in the supplemental material). In contrast to LAERTES-EU data block 2, the remaining bias of block 4 is mostly negative. The highest deviations occur mainly in
mountainous terrain, which may result from the initial resolution differences between E-OBS/LAERTES-EU and HYRAS.

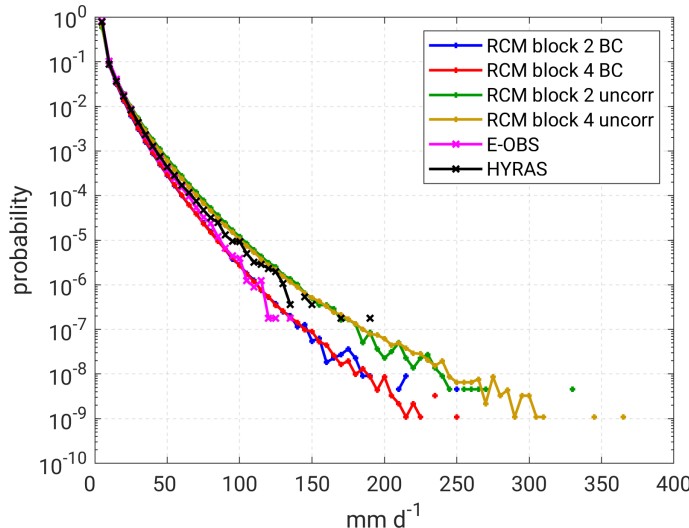

**Figure 3.** Intensity–probability–curve (IPC) of daily rainfall totals within the Rhine basin for LAERTES-EU data blocks 2 and 4, HYRAS, and E-OBS. For LAERTES-EU, the IPCs for the original data set (uncorr) and the bias corrected (BC) data set are shown.

To validate the simulations in a statistical way we use intensity–probability–curves (IPCs). Considering each grid point at each time step, the IPC divides the total range of occurred precipitation values (intensity) into discrete histogram classes and returns their probability. Figure 3 shows the IPCs of LAERTES-EU data blocks 2 and 4 before and after bias correction in comparison with those of E-OBS and HYRAS. After bias correction, the IPCs of LAERTES-EU are in good agreement with the E-OBS curve, but retaining the heavy tail of the distribution, which corresponds to not yet observed precipitation totals. Again an underestimation of E-OBS compared to HYRAS is visible.

The annual cycle of spatially averaged monthly mean precipitation sums (Fig. 4) shows maxima in summer and winter (in agreement with, e.g., Bosshard et al., 2014). Compared to E-OBS and HYRAS, which show similar values, the course of the annual cycle was already well captured in the uncorrected LAERTES-EU data block 2 but with an enhanced amplitude. However, there is a distinct positive bias for all months. Without bias correction, LAERTES-EU data block 4 fails to capture the summer maximum. Instead, a local maximum of precipitation is observed during the spring month. After correcting, the bias is significantly reduced preserving the annual cycle of precipitation. For LAERTES-EU data block 4, the bias correction leads to a stronger reduction in winter and an increase in summer.

From the presented results we conclude that the bias correction provides a clear added value for precipitation fields, distributions, and the annual cycle.

## 4 Hydrological modeling

In this section, we first introduce the used HM. The ability of the HM to simulate extreme discharges is tested by (a) a comparison of observed and simulated discharges in general and (b) for a number of selected historical Rhine river floods.

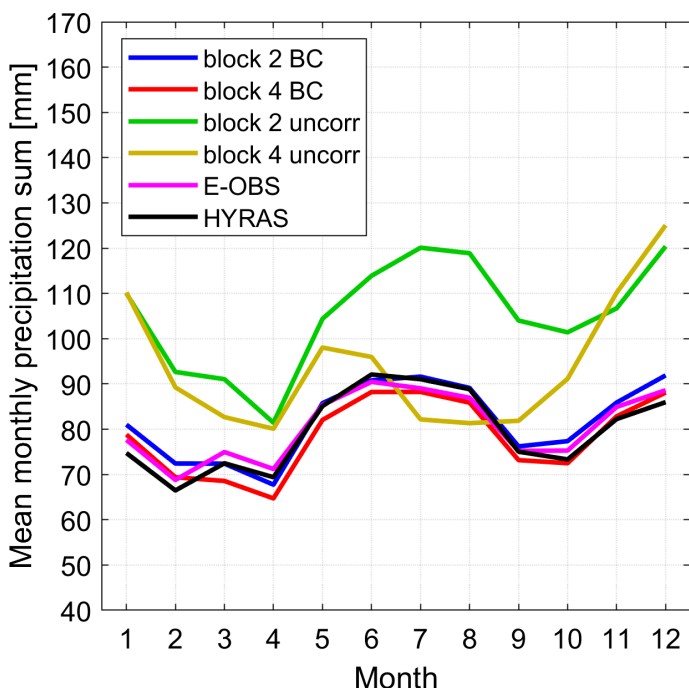

**Figure 4.** Annual cycle of the spatially averaged mean monthly precipitation sum [in mm] based on LAERTES-EU data block 2 and 4 for uncorrected model data (uncorr), bias-corrected data (BC), E-OBS, and HYRAS.

## 4.1 The HBV model approach

The HM used in this study is based on the Hydrologiska Byråns Vattenbalansavdelning model (HBV; Bergström and Forsman, 1973; Lindström et al., 1997). The HBV is a conceptual HM that has been widely used in various hydrological applications ranging from flood forecasting to climate impact assessment (e.g., Lidén and Harlin, 2000; Hunducha and Bardossy, 2004; Olsson and Lindström, 2008; Van Pelt et al., 2009; Arheimer et al., 2011; Cloke et al., 2013; Beck et al., 2013, 2016; Demirel et al., 2015; Vetter et al., 2015; Jenicek et al., 2018; He et al., 2020). Many versions of the HBV model currently exist.

The one used here is based on the HBV-IWS model (He et al., 2011) and has been adapted for spatially distributed input data. It consists of four main routines: (i) snow melt and snow accumulation; (ii) soil moisture and effective precipitation; (iii) evapotranspiration (ET); and (iv) runoff response. A triangular weighting function is used to simulate surface routing delays. Finally, the Muskingum routing method (Cunge, 1969) is used to route the flow from upstream to downstream. The model parameters are calibrated towards observations for each catchment, respectively (He et al., 2011). The model runs at a

daily time step with 5 km grid spacing and requires inputs of daily precipitation, temperature and ET. Since ET is not directly provided by LAERTES-EU, it is calculated from the mean daily temperature following the approach of Oudin et al. (2005). The model was calibrated and validated using the time series of the 71 gauging stations (cf. Sect. 2.2.3). Therefore, the investigation

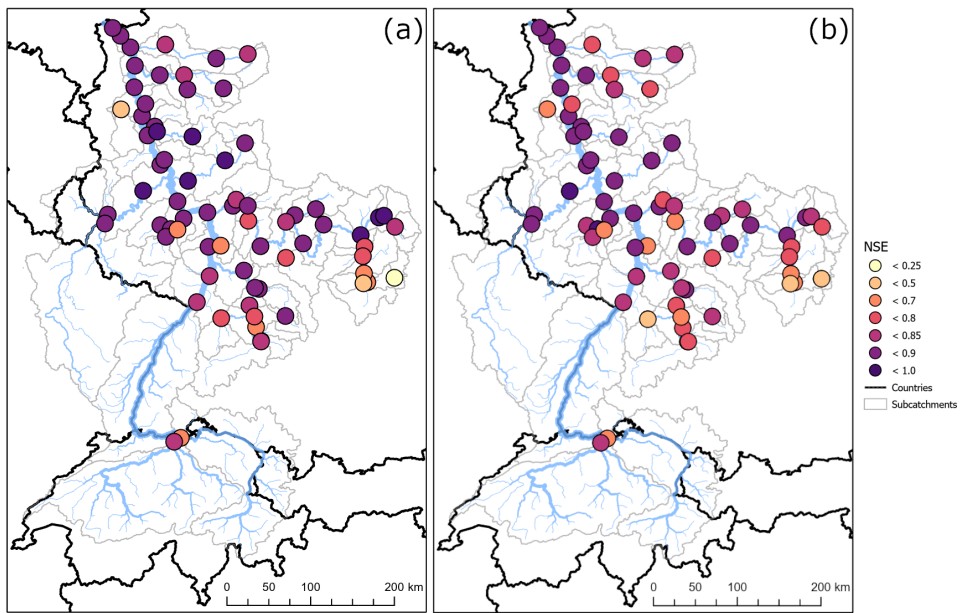

**Figure 5.** Nash–Sutcliffe model efficiency coefficient (NSE) for the 71 catchments of the Rhine basin (validation period 1986–2009) with (a) HYRAS, and (b) E-OBS as HM forcing. Higher values indicate better agreement.

period is split into a calibration part and a validation part. Due to the data availability of all stations, the calibration period is 1975–1985 (11 years) and the validation period is 1986–2009 (24 years). All results are presented for the validation period.

## 4.2 Validation of the HM

### 4.2.1 Discharge representation

In this study, the Nash–Sutcliffe model efficiency coefficient (NSE, Eq. 2; Nash and Sutcliffe, 1970) is used for validating the HBV model. The NSE is a measure of how the simulated discharges match with the observed ones during the validation period. Possible values range between $(-\infty; 1]$ with higher values representing a better match. NSE $= 1$ represents a perfect match between the observation and simulation. The NSE is defined by:

$$\mathrm{NSE} = 1 - \frac{\sum_{i=1}^{N} \left(Q_{i,\mathrm{obs}} - Q_{i,\mathrm{mod}}\right)^2}{\sum_{i=1}^{N} \left(Q_{i,\mathrm{obs}} - \overline{Q}_{\mathrm{obs}}\right)^2} \ , \tag{2}$$

with the observed discharge $Q_{i,\mathrm{obs}}$ at gauge $i$, the corresponding simulated discharge $Q_{i,\mathrm{mod}}$, the mean of all observations $\overline{Q}_{\mathrm{obs}}$, and the total number of considered observations $N$. If NSE $= 1$, the model in the mean is assumed to be unbiased (numerator/sum of deviations equal zero), in case of NSE $= 0$, the predictive skill of the model is as good as the mean of the observations (Krause et al., 2005; McCuen et al., 2006).

The NSE for the 71 individual catchments of the Rhine basin (cf. Sect. 2.3) is shown in Figure 5 for HYRAS (Fig. 5a), and E-OBS (Fig. 5b) as HM forcing. In both cases the NSE shows a good general agreement between the observed and simulated discharges. In fact, only a few of the smaller catchments have a lower NSE. Nevertheless, it also illustrates a better match for HYRAS, which has a higher spatial resolution. As LAERTES-EU is bias-corrected towards E-OBS (due to its spatial availability for entire Europe), we expect the discharge errors caused by the HM to be in the same order, even assuming a perfect precipitation input.

### 4.2.2 Historical flood events

Additionally to the overall performance in the previous section, we analyze in detail three major Rhine river flood events within the validation period: March 1988, December 1993, and January 1995. The time series of simulated and observed discharges are shown exemplary for the Emmerich (EMME) station (cf. Table 2) in Fig. 6. The results for the other gauging stations can be found in the supplemental material (Fig. S2–S6). For those selected case studies, the model is capable to identify flood peaks in terms of timing and intensity. One limitation of the model is to capture significant day–to–day variations in discharge (BETZ, GROL, and ROCK for January 1995), which would require a higher temporal resolution of the HM than daily time steps. A second limitation is the overestimation of flood peaks at EMME of 10–20%, which is likely due to the relatively simple flood wave routing procedure.

## 5 Added value of bias-corrected LAERTES-EU for HM forcing

In the previous section we have provided evidence that the used HM is capable to simulate realistic discharges on a daily basis for different (sub-) catchment extensions. However, the results indicate that a proper representation of input precipitation is beneficial due to the high model sensitivity. We now analyze in how far LAERTES-EU can provide a stochastic data set to represent the statistical properties of observed river discharges.

As LAERTES-EU (both uncorrected and bias corrected) includes simulated precipitation data for thousands of years, we can calculate discharges for different return periods (RPs) from a sorted series of the yearly maximums using the plotting positions approach of Weibull (Makkonen, 2006). For the historical discharges, we have just about 50 years of measured discharges, and 68 (34) years of simulated discharges based on E-OBS (HYRAS). To estimate higher return periods, we need to make assumptions on the underlying distribution of discharge extremes. Although various distributions are used in hydrology, we mainly use a Weibull distribution fitted by the L-moments method (Hosking, 1990) in this study. To illustrate the uncertainty in the distribution selection, we also use Gamma and Gumbel distributions for the observed discharges.

Discharge values derived from LAERTES-EU should have similar distributions of river flow extremes. Figure 7 shows exemplary the distributions of discharge extremes for the EMME station as described in Sect. 2.3 for return periods of 2–2,000 years. The results for the other gauges can be found in the supplementary material (Fig. S7–S11). There are conceptually two different kinds of distributions shown: parametric distributions for short time series (discharge observations, HYRAS, E-OBS) and empirical distributions for long time series (LAERTES-EU). Parametric distribution never perfectly match the data

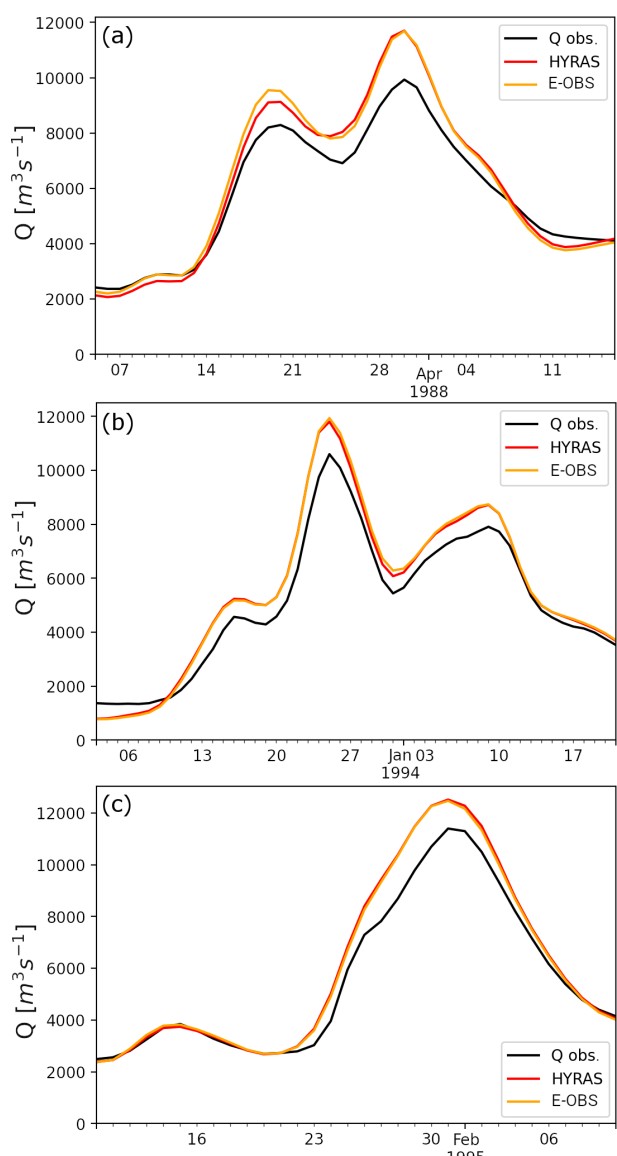

**Figure 6.** Time series of simulated and observed discharges (black) at the Emmerich station (EMME) for the flood events (a) March 1988, (b) December 1993, and (c) January 1995. The simulations are forced with HYRAS (red), and E-OBS (yellow), respectively.

and thus, there is an uncertainty in fitting the distribution parameters which can be visualized as confidence interval (CI) by bootstrapping. The empirical distributions show all data values so there is no mismatch between distribution and data. Figure 7

shows the 95% CI for two parametric distributions: Q Obs-Weibull and HYRAS-Weibull. The Q Obs-Weibull CI represents the uncertainty of the prediction of extreme discharges based on observations (no HM used), while the HYRAS-Weibull CI

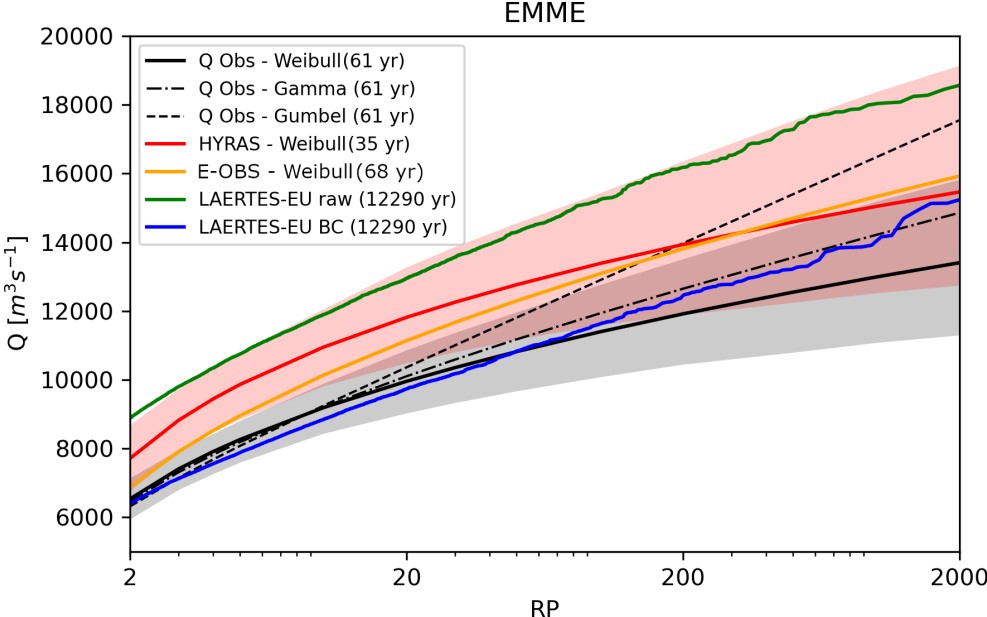

**Figure 7.** Return values of observed and simulated discharges Q at EMME station. Given are the Weibull (black solid), Gumbel (black dashed), and Gamma distributions (black dot-dashed) for observed discharges as well as the Weibull distributions for the simulation forced with observed precipitation from E-OBS (orange) and HYRAS (red). The results from uncorrected LAERTES-EU driven simulations are given in green and those driven by corrected LAERTES-EU data are shown in blue. The shaded areas represent the 95% confidence intervals of HYRAS (red) and Q obs Weibull (gray). The length of each time series is given in the legend.

represents the uncertainty of discharge extremes modeled with the presented HM using the highest resolved data set as forcing which in this case is assumed as best available data.

The results for the two biggest catchments (EMME and FRAN) reveal a clear advantage of using the bias corrected data. LAERTES-EU shows better results than the simulations driven with observed precipitation from E-OBS or HYRAS. The distribution of return periods estimated from LAERTES-EU is in good agreement with those of the observations, especially when considering different distribution functions and CI. At the ROCK station, the uncorrected discharge extremes are too high and outside the CI of the observations. These extreme values are reduced by the bias correction, but the reduction also leads to an underestimation of lower return periods, where the historical data are reliable. For the smaller catchments, the effect of bias correction is mixed (sometimes positive, sometimes negative). As LAERTES-EU was corrected towards E-OBS, the bias correction has a negative impact. This is particularly true in medium and small catchments where E-OBS shows significant differences to HYRAS in terms of precipitation. At the GROL station, the discharges forced by historical E-OBS precipitation are underestimated and thus, the bias-corrected stochastic discharges are also underestimated. Possible reasons are the relatively coarse resolution of LAERTES-EU and the daily time resolution. Both facts prevent to capture small-scale and/or convective phenomena with short duration. Nevertheless, smaller catchments (and rivers) show a higher sensitivity for such events.

**Table 3.** Return interval $T_{obs}$ of the peak discharge Q for selected historical flood events and stations (sorted in descending order regarding catchment size, cf. Table 2) estimated from the Q obs-Weibull distribution and 95% CI ($T_{0.05}$ and $T_{0.95}$) and bias-corrected LAERTES-EU data $T_{LAERTES}$.

| Event | station | Q [$m^3s^{-1}$] | $T_{obs}$ [yrs] | $T_{0.05}$ [yrs] | $T_{0.95}$ [yrs] | $T_{LAERTES}$ [yrs] |
|-------|---------|-----------------|-----------------|------------------|------------------|---------------------|
| Mar 1988 | EMME | 9930 | 17 | 10 | 81 | 24 |
| | FRAN | 1760 | 26 | 13 | 206 | 30 |
| | ROCK | 1810 | 14 | 8 | 36 | 37 |
| | GROL | 291 | <2 | <2 | 3 | 5 |
| | BADV | 62 | 4 | 3 | 5 | 5 |
| | BETZ | 145 | <2 | <2 | 2 | 6 |
| Dec 1993 | EMME | 10600 | 30 | 15 | 313 | 42 |
| | FRAN | 1220 | 5 | 3 | 8 | 5 |
| | ROCK | 2140 | 37 | 16 | 188 | 110 |
| | GROL | 727 | 55 | 14 | 810 | 1855 |
| | BADV | 80 | 15 | 9 | 76 | 26 |
| | BETZ | 214 | 5 | 3 | 8 | 36 |
| Jan 1995 | EMME | 11400 | 61 | 29 | 3590 | 82 |
| | FRAN | 1990 | 57 | 23 | 1245 | 66 |
| | ROCK | 1130 | 3 | <2 | 4 | 4 |
| | GROL | 689 | 38 | 12 | 420 | 734 |
| | BADV | 81 | 17 | 10 | 103 | 30 |
| | BETZ | 222 | 6 | 4 | 10 | 46 |

The results presented in Fig. 7 and Figures S7–S11 are used to estimate the return interval of the historical flood events used for the model validation in Sect. 4.2.2 and presented in Table 3. Similar to the figures, the results in Table 3 show a rather large uncertainty range in most of the cases. For the biggest two catchments (EMME and FRAN), the return periods from LAERTES-EU are close to those estimate from observed discharges and within the CI of the observations but closer to the lower CI boundary. This indicates that the observed estimates tend to overestimate return periods. For the medium size catchment represented by ROCK station, the LAERTES-EU return periods are in the range of the upper observed CI boundary or slightly above. For the smaller catchments (GROL, BETZ, and BADV) the results are mixed but LAERTES-EU tends to overestimate the return periods. Again, possible reasons are the spatial and temporal resolution of LAERTES-EU which might not be sufficient to capture flood events in these catchments.

## 6 Summary and Conclusions

In this study, we have adapted, applied, and validated the LAERTES-EU precipitation data set for hydrological applications in the Rhine river basin. The main aims were to reduce the positive precipitation bias of LAERTES-EU already stated by Ehmele et al. (2020) compared to meteorological observations and to improve hydrological discharge simulations with respect to a more robust statistical representation of extremes characterized by high return periods. Following the formulated research questions (Sect. 1), the main conclusions are as follows:

1. The mainly positive precipitation bias of the original LAERTES-EU data was reduced to a large extend by the bias correction approach. The statistical distribution of precipitation now follows that of the observations but conserves the heavy tail representing not yet observed (extreme) values. The typical characteristics like the annual cycle are conserved but improved in terms of amplitude.

2. The applied HM can reproduce historical flood events in terms of peak discharge and timing. Nevertheless, the results are case sensitive and depend on the catchment size and related terrain characteristics. Moreover, the results demonstrate the necessity of a proper representation of the forcing data.

3. Discharge simulations for the Rhine basin demonstrate a proper representation of discharge distributions even for high return periods using LAERTES-EU as input data due to the large ensemble size. The bias-corrected precipitation input improves the representation of discharge return values within the uncertainty of observations associated with the extrapolation required for return periods beyond the observed record. Nevertheless, the results depend on the catchment size.

Regarding (1), we provide evidence that the applied bias correction works properly across the whole model chain. The positive precipitation bias of LAERTES-EU (Ehmele et al., 2020) is reduced to a large degree. The statistical distributions like IPCs and the annual cycle are now in good agreement with the E-OBS reference data. The applied methodology of adaptive correction functions (depending on data block, month) has many advantages. For example, it enables the consideration of different bias magnitudes across the year, with a stronger adjustment during winter months. Treating a LAERTES-EU data block as a single data set, the internal variability of the single members within a data block is conserved. Furthermore, the approach retains the heavy tail of the distribution representing the not yet observed range of values, as can be expected from such a long data set. However, the quality of the bias correction strongly depends on the reference data set and is therefore limited to the quality of observations. Following Haas et al. (2014) for wind speed, a sparse data density or availability can worsen the results after bias correction. In case of the Rhine basin, the data density is quite high and thus, the bias correction is expected to have a good quality. The IPCs point out that E-OBS has a negative bias compared to the higher resolved HYRAS data set, and thus, the corrected LAERTES-EU also shows a negative bias towards HYRAS. However, it is out of the scope of this study whether E-OBS or HYRAS is qualitatively better for the bias correction in case of the Rhine basin. Apart from this, a broader context has to be considered as the main aim is a bias correction on the entire EURO-CORDEX domain for which only E-OBS has sufficiently long time series.

Regarding (2), we have applied the HM to historical flooding events (three cases for the Rhine) using observations as forcing. We provide evidence that the HM can reproduce these events properly in terms of timing and peak discharge. Deviations to observed discharges can be attributed to some limitations of the used data sets and HM, like the relatively coarse spatial resolution and the daily time step. The former has mainly a significant impact in mountainous terrain or for small catchments while the latter mainly affects the flood wave propagation and timing. Nevertheless, a timing error is identified in a few cases and magnitude deviations can be further post-processed.

Regarding (3), the quality of the discharge simulations strongly depends on the catchment size. For the entire basins or large catchments, the bias correction clearly has an added value, given that the estimated discharge return periods are remarkably close to the observations which were extrapolated for high return periods using several distribution functions. The uncorrected data leads to a general overestimation of discharges. For smaller catchments, the results are more mixed. In cases where E-OBS driven simulations show low discharges, the simulations after bias correction also show an underestimation. This behavior can be explained with the stronger sensitivity of the smaller catchments to small-scale and/or convective phenomena as well as sub-daily effects (e.g., Seibert and Auerswald, 2020). Due to the limited length of observational records, the estimated return values for high return periods show a high uncertainty. From a statistical point of view, the large amount of data of LAERTES-EU should enable more robust estimates in that context at least for large and medium size catchments as seen in the estimated return periods for selected historical flood events.

LAERTES-EU consists of various simulations of a single RCM (CCLM) downscaling different realizations at different resolutions of a single GCM (MPI-ESM). As the RCM is identical for all simulations, differences mainly originate from the forcing GCM data and the internal variability. Therefore, a proper representation of the typical weather patterns associated with floods over Europe in the GCM is important. As pointed out, e.g., by Cannon (2020), the used MPI-ESM shows a good representation of the typical weather patterns for the European sector. Comparable results should be achieved using a GCM with similar quality and climate sensitivity. Stronger diverging results are expected to emerge only for GCMs with a different representation of the regional climate and a stronger/weaker climate sensitivity.

The LAERTES-EU precipitation data was bias-corrected applying the quantile mapping technique with a parameterized gamma distribution. Although the LAERTES-EU data follow the extrapolated discharge observations (also using a gamma distribution), the choice of parametric function for the bias correction has only little or no influence on the statistical distribution of discharges due to the high non-linearity in the used HM. If a catchment consists of only one precipitation grid cell or in small mountainous catchment, where most of the rain quickly triggers discharge, the linkage could be stronger. For bigger catchments or the entire Rhine basin, the overall precipitation sum over the area and also the timing play a much bigger role.

Beside the presented benefits, there are some limitations of the data set. LAERTES-EU is not expected to reproduce specific historical events in a deterministic way (like a short-term weather forecast for a maximum of 10–14 days) but probabilistic with associated uncertainties. Another limitation is the comparatively coarse resolution for impact studies of approx. 25 km, which causes a distinct bias especially in strongly structured mountainous terrain. Despite these shortcomings, LAERTES-EU provides robust spatially and time consistent stochastic precipitation data to estimate even high return levels. Another advantage is the provision of a multivariate and to a large degree consistent precipitation and temperature data set, which is necessary to

also consider the effects of snow accumulation and snow melt. The applied HM uses daily mean temperature to decide whether precipitation should be treated as rain or snow. Furthermore, temperature is used to calculate additional input variables like ET.

Although the physical relation between precipitation and temperature might be changed after the bias correction, the large-scale dynamics that produce specific weather patterns and precipitation fields remain largely unchanged. Terink et al. (2010) argued that the correlation between temperature and precipitation is rather small for the Rhine basin. The comprehensive analysis of frequency and characteristics of precipitation and flooding events for a wide range of return periods under present climate conditions is thus possible. Given the spatial and temporal consistency of the data set, it is possible to investigate flood events

that take place in multiple basins at the same time.

The resulting methodology and obtained discharge data can be used to develop probabilistic catastrophe models and risk assessments. This can be performed not only for single catchments but on national and pan-European scales, combining the extreme value statistics from multiple river basins. In particular, adaptations and applications of the presented methodology are ongoing for several large Central European river basins such as the Danube, Elbe, Oder, or Vistula basin. Regarding

hydrology, some recalibration of the HM set up to further improve the model performance in these basins is ongoing. For instance, the results can be post-processed (scaled) for further impact modeling using a quantile-quantile mapping technique. This calibration step will fix the underestimation of peak discharge values while maintaining the large spatial and temporal variability of simulated floods from LAERTES-EU. Regarding the atmospheric part, LAERTES-EU will be used in a follow-up study investigating the relation between the spatial variability of precipitation over Europe and teleconnection patterns. Further

applications of LAERTES-EU can include a statistical and/or combined statistical-dynamical downscaling towards higher resolutions to improve both precipitation and discharge representation, especially over mountain ranges. Other extensions could be the evaluation of other variables/hazards and the investigation of so-called compound events, i.e., simultaneously occurring multiple hazards (e.g., Zscheischler et al., 2018; Raymond et al., 2020). The analysis can also be extended by considering climate projection scenarios (e.g., RCP4.5/RCP8.5; Jacob et al., 2014) to estimate possible changes in the frequency, intensity,

and extension of hydrometeorological extremes in the 21[st] century.

*Data availability.* E-OBS (Haylock et al., 2008) is available after registration at https://www.ecad.eu/download/ensembles/ensembles.php (last access: 2 December 2020). HYRAS (Rauthe et al., 2013) can be requested at the German Weather Service (DWD). LAERTES-EU (MiKlip data) will be made available via the CERA database (http://cera-www.dkrz.de/; last access: 2 December 2020) of the German Climate Computing Center (DKRZ). Discharge observations can be requested from the respective competent authorities (cf. Sect. 2.2.3).

*Author contributions.* FE and LAK contributed equally to this study. FE, LAK, HF, and JGP designed the study. HF performed (parts of) the RCM simulations. FE did the bias correction, some analyses and wrote the initial draft. FDK contributed to the bias correction with programming expertise. YH and DM developed the hydrological model and wrote the model description. LAK and HSL contributed some precipitation analyses. MK runs the hydrological model and made the corresponding analysis. All authors contributed with discussions and revisions.

*Competing interests.* Joaquim Pinto is editor for NHESS. The other authors declare that they have no conflict of interest.

*Acknowledgements.* We acknowledge the E-OBS data set from the EU-FP6 project ENSEMBLES (http://ensembles-eu.metoffice.com, last access: 18 December 2020) and the data providers in the ECA&D project (https://www.ecad.eu, last access: 18 December 2020). We also thank the German Weather Service (DWD) for providing HYRAS. In addition, we thank the German Climate Computing Centre (DKRZ, Hamburg) for computing and storage resources in project 983. We also thank the Zentralanstalt für Meteorologie und Geodynamik (ZAMG) for providing SPARTACUS and all national and regional competent authorities for providing discharge observations. We thank AON for funding the project "Hydrometeorological extreme events under recent climate conditions". We also thank the BMBF MiKlip project II (FKZ: 01 LP 1518 A/D) and ClimXtreme Project (FKZ: 01 LP 1901 A) for partial funding. Patrick Ludwig was partially funded by the Helmholtz Climate Initiative REKLIM (regional climate change; https://www.reklim.de/en, last access: 18 December 2020). Joaquim G. Pinto thanks the AXA Research Fund for support (https://axa-research.org/en/project/joaquim-pinto, last access: 18 December 2020). We thank open-access publishing fund of the Karlsruhe Institute of Technology (KIT). We thank the reviewers for their valuable comments that helped to improve this study, and the handling editor for guidance throughout the entire process.

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
