# Peer review of "Adaptation and Application of the large LAERTES-EU RCM Ensemble for Modeling Hydrological Extremes: A pilot study for the Rhine basin"

_Natural Hazards and Earth System Sciences, 2021_

## Author Response (AR1)

**List of Major Changes**

1) We updated Section 2.1 on the LAERTES-EU data set to provide more detailed information.
2) We added the contour of the Rhine basin to Figures 2 and S1.
3) For the validation of the used hydrological model (HBV, Section 4.2), we split the investigation period in a calibration part (1975-1985) and a validation part (1986-2009) and now show results for the validation period, only.
4) We updated Figures 7 and S7-S11 adding the 95% confidence intervals of observed discharge distribution (Q obs Weibull) and the distribution of simulated discharges with HYRAS precipitation as forcing.
5) We add a new Table (Table 3) presenting and comparing return periods for the historical flood events used in the validation section estimated from observed discharge extrapolations and from LAERTES-EU. A respective text paragraph was added, too.
6) All text passages are updated accordingly.
7) Additional remarks are added to the conclusions.

**Point-by-point response to Reviewer #1**

Florian Ehmele on behalf of all co-authors
Updated December 20, 2021

**Dear Reviewer No. 1, Thank you very much again for your work and the useful and valuable comments that will help to improve the scientific quality of our manuscript. Below you will find your comments given in gray and our updated responses to the individual points in black. Please also consider our comments to Reviewer 2 as there is some coincidence of the comments and the corresponding answers.**

*This is an interesting paper that describes the use of a large ensemble of regionally downscaled multi-GCM forcings to drive a hydrological model for impact assessments. The issue of long return period extremes is highly relevant. The paper is very well written, clearly structured and to the point. However, there are some unfortunate shortcuts regarding the model validation which needs to be handled differently.*

**Thank you very much. We hope to implement your comments in a sufficient way.**

*Main comments:*
*Both the bias correction and the HBV set ups are validated on the calibration period. While I can accept this for the bias adjustment because it is not anywhere applied outside of the calibration period, it is a big issue for the justification of the hydrological model. HBV is currently calibrated and validated on the same period (1961-2006) based on precipitation and temperatur forcing from gridded observational data sets. When validated on that same period, the results are very good, as seen from the very high NSE values. However, we still know nothing about the model's performance on data it has never seen before, and the main results are based on the downscaled model data. I urge the authors to at least perform a split sample validation where calibration and validation periods are independent, or even a cross-validation. This is standard practice in hydrological model validation.*

**We agree with the reviewer that proper practice is a split between calibration and validation period in case of the HBV model. For the revised version we have split the considered time period into a calibration (1975-1985) and a validation phase (1986-2009) and present the NSE results for the new validation period only. As NSE is a robust statistic, only small changes arise.**

*Bias correction is only performed for precipitation, and no information about potential bias in temperature and how it might affect results is provided. Because temperature, and its translation into evapotranspiration, is an important input to the water balance of the model, it should not be neglected. I would like to at least see a justification for why temperature is not bias corrected (being that the bias is low). In some cases it can be neglected for certain extremes where the pre-conditioning of the river is of minor importance, but also that needs some additional analysis and commenting in the text.*

**We agree that evapotranspiration and therefore also temperature is important for the total water budget. The LAERTES-EU temperature data have also been bias-corrected using the quantile mapping approach with a Gaussian distribution function. The bias-corrected temperature data have been used in line with bias-corrected precipitation. Nevertheless, the dominant factor in case of the major flooding events is precipitation so we focus on the precipitation part of the bias correction. We have added a comment on that.**

*The concluding main result of the paper is presented in figure 7. Although the result is compelling and seemingly clear, the details may occlude the actual results. First, the lenght of each timeseries has a large effect on the GEV fits and their robustness, as argued in the introduction. Please add the record lenght, i.e. the number of years, in the legend for each data set.*

**We agree with the reviewer that the length of the time series has a crucial impact on the estimated distributions. As requested we have added the length of each time series in the legend of the plots and in terms of discharge data also in Table 2.**

*Second, it would help the reader a lot to also see the confidence intervals. With so many lines, it might get too busy, but I think adding e.g. the confidence interval for the "Q obs – Weibul" and "LAERTES-EU BC" would be very informative. The confidence intervals would convey two results, one is the fair comparison of the observations and the model that would show the observations results essentially useless beyond 50 years (depending on the lenght of the timeseries), and the other is the added value of the multi-realization simulations which add statistical robustness for the longer return periods.*

**Thank you for this comment which we totally agree with. For short time series it is necessary to fit distributions to the data to extrapolate discharge at higher return periods and the extrapolation has significant uncertainty. But for LAERTES-EU we have over 12,000 years and need only discharge at RP 2,000 for the application of this project. This can be read visually from the empirical graph, and hence theoretical distribution isn't needed for extrapolation. That is also why the LAERTES-EU lines are curvy. We plotted the sorted modeled discharges (empirical distribution), not a smooth parametric distribution. There is some uncertainty in assigning return periods to the sorted yearly maximum discharges (method called plotting positions) but it is tiny compared to uncertainty of extrapolated data. For a profound estimate of the confidence intervals of an empirical distribution, the dataset has to be split into sub-samples with each subset being long enough for a robust estimation of high return periods up to 2000 years. This would be possible only for a much larger dataset than LAERTES-EU. Nevertheless, we have added the confidence intervals for the parametric distribution estimates of Q obs Weibull and HYRAS in Fig.7 and S7-S11. We have also adjust the text on that. As suggested by the editor we have also included the new Table 3 providing return period estimates for the used historical flood events in the validation section. Table 3 compares the return periods from the Q obs Weibull extrapolation with CI95 and the LAERTES-EU estimates.**

*Minor comments:*
*L69: Please clarify what you mean with "isolate the effects".*

**It was meant to elaborate the changes that the bias correction brings to both precipitation and discharge statistics and the added value for an application such as the one presented in this study. We have rewritten this sentence for clarification.**

*L85-94: Please describe more details about the LEARTES-EU multi-model. It is currently not clear what the driving GCMs are; especially that they area mixture of assimilated reanalysis, decadal initialized forecasts and free GCM simulations. Please repeat more from Ehmele et al. (2020) which provides a good summary, enough for the reader to understand from this paper alone.*

**We have rewritten the section on the LAERTES-EU dataset and provide more information similar to the original study by Ehmele et al. (2020) but still keeping it short and concise.**

*L176, 185: I would avoid describing differences between data sets as "bias", but rather use the word "difference" unless you include a well established ground truth observational reference.*

**We agree that "bias" is misleading in this context. As suggested we have changed to "differences" or other related terms where appropriate.**

*Figure 3: Please consider using a log-log scale, which would better show differences between the data sets for the (0,100) mm/day range.*

**Thank you for this suggestion. We already tried a log-log scale plotting, which is added below. We agree, that in theory the range between 0 and 100 mm can be better recognized in a log-log scale. As shown in the figure, there are only small differences in this intensity range below 50 mm which can also be recognized in the single y-log version. Furthermore, the more interesting and relevant part of the distribution is the heavy tail which is better represented in the single y-log scaling. So we decided to keep Figure 3 as it is.**

[Figure]

*L307: "different forcing and/or assimilation schemes". I refer back to my earlier comment that the LAERTES-EU sources needs to be better described.*

**We have added more information on LAERTES-EU, please refer to the comment above for details.**

*L310: "consistend data for precipitation and temperature". This is not really true after bias correction. The depencency between the variables can be severely impacted. You have also not described the potential temperature bias and how it migh affect rain/snow distribution and timing over the year. It might not be useful to retain the dependence it is errouneous?*

**We agree that "consistent" is potentially misleading in this context and care has to be taken when using it. As mentioned before the dominant factor in case of major flood**

events is precipitation. The dependency disruption is limited so that the data set can be treated as almost consistent or "consistent to a large degree". Furthermore, the large-scale dynamics that produce specific weather patterns and precipitation fields are not influenced by the bias correction so that a synoptic situation leading to heavy precipitation remains the same after bias correction. We have changed the wording in the text accordingly and also added some additional comments on this topic in the conclusions.

*Figure S7-11: Please change "Observed - Weibul" to "Q obs. - Weibul" as in the main text figure.*
This was accidentally forgotten to adjust. We have fixed this in the revised version.

**Point-by-point response to Reviewer #2**

Florian Ehmele on behalf of all co-authors
Updated December 20, 2021

**Dear Reviewer No. 2, Thank you very much again for your work and the useful and valuable comments that will help to improve the scientific quality of our manuscript. Below you will find your comments given in gray and our updated responses to the individual points in black. Please also consider our comments to Reviewer 1 as there is some coincidence of the comments and the corresponding answers.**

*The manuscript by Ehmele et al. investigates the use of a large ensemble of RCM simulations instead of very long observational times series (of precipitation and river discharge) in estimation of return periods. This is very interesting, esp. because of the possibility to use a consistent meteorological dataset in forcing a hydrological model for discharge calculations. Still, I have questions which are detailed below.*

**Thank you very much. We hope to implement your comments and answer your questions in a sufficient way.**

*The approach is successful only after bias correction of the RCM output as is shown in literature and by the authors. The bias correction of precipitation relies on a quantile method applying the Gamma distribution. Does this imply some statistical behavior of the return period derived? It follows quite nicely the observation-based return periods extrapolated assuming the Gamma distribution (in Fig. 7). Asked differently: is there an added value of LAERTES-EU in return period estimation as it must rely on bias-correction using observational data and some statistical assumption?*

**Thank you for this comment. If the catchment consists only of one precipitation cell, then there is strong link between the precipitation distribution and distribution of discharges. However, the rainfall-runoff model is non-linear and the response varies a lot catchment by catchment, so the transformation is not simple. For small mountain catchments where most of rain immediately transforms to discharge the link could be so strong that it may imply a Gamma distribution of discharges as well. But this is not a general relation as for the bigger catchments, the sum of precipitation over the whole catchment and also the timing play a much bigger role than one actual value of precipitation in an individual cell, so the link is much weaker. The added value of using the bias corrected forcing from the LAERTES-EU is to provide more robust estimation of extreme floods of large return periods. This is evident from Figure 7. The estimated flood return periods obtained from LAERTES-EU BC shows the closest match with the Q obs. The uncorrected LAERTES-EU forcing leads to considerable underestimation of the return periods. E-OBS or HYRAS forcing also leads to under-estimation of the return periods because of limited data length, and hence unreliable estimation. We have added some comments on that in the text of the corresponding section and also in the conclusions. Furthermore, we added the new Table 3 as suggested by the editor providing return period estimates from extrapolated observations and LAERTES-EU for the historical flood events used in the validation section.**

*Why is there some precipitation bias in the Alpine area after bias correction (Fig. 2)?*

**Figure 2 shows the ensemble mean of LAERTES-EU data block 2 which consist of multiple decadal simulations (see Sect. 2.1 and Table 1). The gamma distribution used for the quantile mapping bias correction is calibrated using the data block in total and not for every single ensemble member by itself. The main intention was to preserve the natural**

variability of the ensemble. Consequently, there are ensemble members for which the bias correction work well, for some a positive bias will remain and others have a negative bias. In the ensemble mean, a small absolute positive bias of roughly 0.2-0.4mm remains. This is more likely to happen in areas with complex terrain and therefore a higher spatial variability of the precipitation field which is more difficult to capture by RCMs especially at the used resolution of approx. 25km. We have added a comment in the text.

*The intensity-probability curve of the uncorrected RCM precipitation follows nicely the HYRAS near-observation curve and less the E-Obs curve in Fig. 3. If we assume that HYRAS is better in Germany than E-OBs, why can we not conclude that bias-correction deteriorates the probabilities?*

The bias correction always depends on the quality of the used references/observations. As shown, for example, by Haas et al. (2014)[1] for wind speed, a sparse data density or data availability can worsen the results during and after bias correction. For Central Europe, or in particular Germany and the Rhine Basin, the data density is quite high, so we expect also a high quality of the bias correction. We totally agree that better results can be expected for Germany using HYRAS as reference due to the higher spatial resolution and the underlying number of stations HYRAS used for the interpolation. Nevertheless, it is out of the scope of this study whether E-OBS or HYRAS is qualitatively better for the bias correction in case of the Rhine basin and a broader context has to be taken into account as the main aim of the project is to bias-correct LAERTES-EU for a much larger domain that goes beyond the Rhine basin where we would need to reply on E-OBS only. The Rhine Basin in this study served as a pilot area. The overall domain of LAERTES-EU is the EURO-CORDEX domain and the overall aim is a bias correction on that large region. Therefore, a reference covering the entire EURO-CORDEX domain and for a sufficiently long time period was needed which is E-OBS. We have included some remarks on this in the revised manuscript.

[1]*Haas, R., Pinto, J. G., and Born, K. (2014), Can dynamically downscaled windstorm footprints be improved by observations through a probabilistic approach?, J. Geophys. Res. Atmos., 119, 713– 725, doi:*10.1002/2013JD020882*.*

*LAERTES-EU downscales different MPI-ESM GCM versions and ensembles. Still, how important is the imprint of MPI-ESM on the representation on extremes? Can we expect substantially different return periods fi using a different GCM?*

The differences between the blocks of simulations are in fact quite small (cf. Ehmele et al., 2020 Fig. 2 and Table 2). For LAERTES-EU the setup of the RCM (COSMO-CLM) remains the same in all simulations. The forcing GCM is MPI-ESM in different resolutions (high = HR; low = LR) and with different emission protocols (CMIP5 and CMIP6). However, both are applied mainly over the historical period with observed greenhouse gas forcing. The most important aspect w.r.t. the choice of the forcing GCM is a good representation of the relevant weather pattern in the region of interest. Several studies show such a good representation for the European sector as well for MPI-ESM-LR and MPI-ESM-HR (e.g. Cannon et al., 2020)[2]. We would expect comparable results from using a GCM with a similar quality and climate sensitivity and larger differnces only for GCMs with a poor representation of the regional climate and a significantly stronger/weaker climate sensitivity. We have added comments on this in the conclusions.

[2]*Cannon 2020: Reductions in daily continental-scale atmospheric circulation biases between generations of global climate models: CMIP5 to CMIP6 Environ. Res. Lett. 15 064006,* https://doi.org/10.1088/1748-9326/ab7e4f

*Line 8: What means "fixed" here?*

**"Fixed" in this context means, that we used the quantile mapping technique with a parameterized probability density function, here a two-parameter gamma distribution. The alternative way would be an empirical distribution which would strongly follow the observed/modeled values with a significant impact not only on the shape of the distribution but also for the range of values. We explicitly decided to use a parameterized function to preserved to a certain extend the heavy tail of the precipitation distribution which represent the unknown or not yet observed events which would have been corrected using an empirical approach. We have rewritten this sentence for clarification.**

*Tab. 1: block 3: EMS -> ESM, block 4 is given two times, and why not using the new CMIP6 ensemble?*

**We have fixed the typo in Table 1. In alignment with Ehmele et al. (2020), data block 4 is divided into two sub-parts. Both parts consist of decadal simulations and the driving model for both parts is the MPI-ESM-HR. For the first part, the CMIP5 based simulations are used and for the second part the newer CMIP6. As the driving model (MPI-ESM-HR) and the methodology (decadal simulations) are the same, these two parts concatenated to one block of LAERTES-EU. All the simulations within LAERTES-EU have been performed within the BMBF (German ministry of education and research) project MiKlip (medium-range climate predictions) which ended in 2019, where CMIP6 had limited availability. As requested by Reviewer #1, we have added some more information on LAERTES-EU from the original Ehmele et al. (2020) study in the revised manuscript which hopefully make it clearer.**